# Brief empathy interventions online can decrease but not increase empathic tendencies
Alexander Tagesson [1,2] ✉, Annika Wallin [1] & Philip Pärnamets [3]

People often feel less empathy towards outgroup compared to ingroup targets. Overcoming this intergroup empathy bias is important for fostering positive intergroup relations. In five pre-registered and high-powered online studies (n = 4776 (745/745/1056/1236/994)), we attempted to replicate and generalize motivated empathy interventions that previously have made people more empathetic and prosocial towards outgroup targets. Using both between- and within-subject designs, self-reported empathy measures and factual monetary donations, we examined the effects of several brief interventions. The interventions targeted avoidance motivations based on beliefs about the un/limited nature of empathy or approach motivations based on beliefs about empathy's malleability or normatively desirability. Across studies, we tested the interventions in several in- and intergroup contexts, using both novel and preexisting stimuli. In general, interventions failed to increase empathy or prosocial behaviour. Instead, inducing beliefs about the limited nature of empathy often reduced participants' empathy. Motivating people to withhold empathy may be easier than motivating them to increase it.

Empathy is often thought to positively affect social cohesion and care for others[1]. For example, empathy drives prosocial behaviour[2], and empathic people have more friends[3] and more relationship satisfaction[4]. However, we tend to empathize more with people belonging to the same group as we do, and less with those affiliated with other groups[5–9]. To create a better understanding of others and mitigate tension between different groups, empathy researchers try to breach this "intergroup empathy bias."[10–16]. Recent work has targeted people's motivation to feel empathy to counteract intergroup empathy bias[17,18]. One practical challenge with existing interventions is that they require considerable time and effort to implement. If shorter, efficient interventions are found this will increase research's potential to solve real-world problems. One such brief intervention has shown great promise for generating empathy with outgroup members. In a recent paper, Hasson and colleagues were able to increase the empathy participants expressed toward members of other social groups by informing them that empathy was an unlimited rather than a limited resource[18]. This is based on an understanding of empathy as being a malleable process and that people can become more empathic if they are motivated[1]. Here we attempted to generalize and replicate Hasson and colleagues' findings as well as test other brief interventions which we modelled on existing longer motivation-based ones.

Empathy is generally understood as consisting of an affective component: experience sharing; a cognitive component: mentalizing; and a motivational component, referred to as empathic concern[19–21]. Individuals have different propensities to feel empathy[22] and dynamically up and downregulate it to fit their goals[23,24]. This regulation is commonly framed as a motivated process[1,25], where, for example, a desire for affiliation motivates individuals to feel empathy and competition encourages individuals to avoid it. People's shifting motivations can partially explain why people readily feel empathy in some situations but avoid it in others, such as in intergroup contexts.

Considerable research has found that people are more likely to empathize with ingroup over outgroup targets[5,8,9,13,18,26]. One explanation for this comes from Social Identity Theory[27–29]. According to Social Identity Theory, people derive self-worth from the social groups they belong to, and the values and emotions associated with being a member of those groups. Self-categorization into groups is largely automatic and has wide-ranging effects on cognition and behaviour[30–33]. Social groups can be based on different and sometimes arbitrary criteria ranging from favourite sports teams to ethnicity and political affiliation[27,34,35]. Intergroup tensions are an unfortunate effect of this social categorization, which contributes to prejudice, discrimination, and violence towards outgroup members[36,37] These tensions are partially explained by reduced empathy towards outgroup

[1]Department of Philosophy, Division of Cognitive Science, Lund University, Lund, Sweden. [2]Agenda 2030 Graduate School, Lund University, Lund, Sweden. [3]Division of Psychology, Department of Clinical Neuroscience, Karolinska Institute, Stockholm, Sweden. ✉e-mail: alexander.tagesson@lucs.lu.se

targets, especially since empathy often implies both material[38,39] and cognitive cost[40,41].

People value empathy and associate it with desirable traits, such as kindness and generosity[1,17]. Desirable resources are often perceived as limited[42], and people tend to perceive empathy in itself as a restricted resource[6,18,43–45]. Hasson and colleagues suggest that people want to keep scarce empathic resources for their ingroup, and consequently feel less empathy toward outgroup members[18]. Accordingly, they manipulated participants' beliefs about empathy and found that participants induced to think of empathy as an unlimited resource were more likely to feel empathy with outgroup members compared to participants induced to consider empathy as a limited resource. A particularly impressive feature of Hasson and colleagues' studies is that they used a brief intervention consisting of only a few sentences and a scale rating (their studies 2 & 3, see Fig. 1B) to

motivate empathy with outgroup members. Despite its brevity, this brief intervention produced a medium effect size when participants' empathic reactions toward outgroup members were compared between the limited and unlimited conditions (study 2 & 3). If replicable, these results testify to the impact brief motivational interventions can have.

Other successful, but longer, interventions targeting empathy motivations have been reported in the literature. Weisz and colleagues[17] utilized a motivated empathy approach to increase college freshmen's empathy. The interventions consisted of three sessions, approximately one hour each, where participants learned about empathy, including reading, writing, and practicing a speech about it. Some participants learned that empathy is malleable and that they could become more empathic by adopting the right mindset. Others learned that empathy is socially desirable and normatively appropriate[14,46]. Eight weeks after completing the interventions, participants

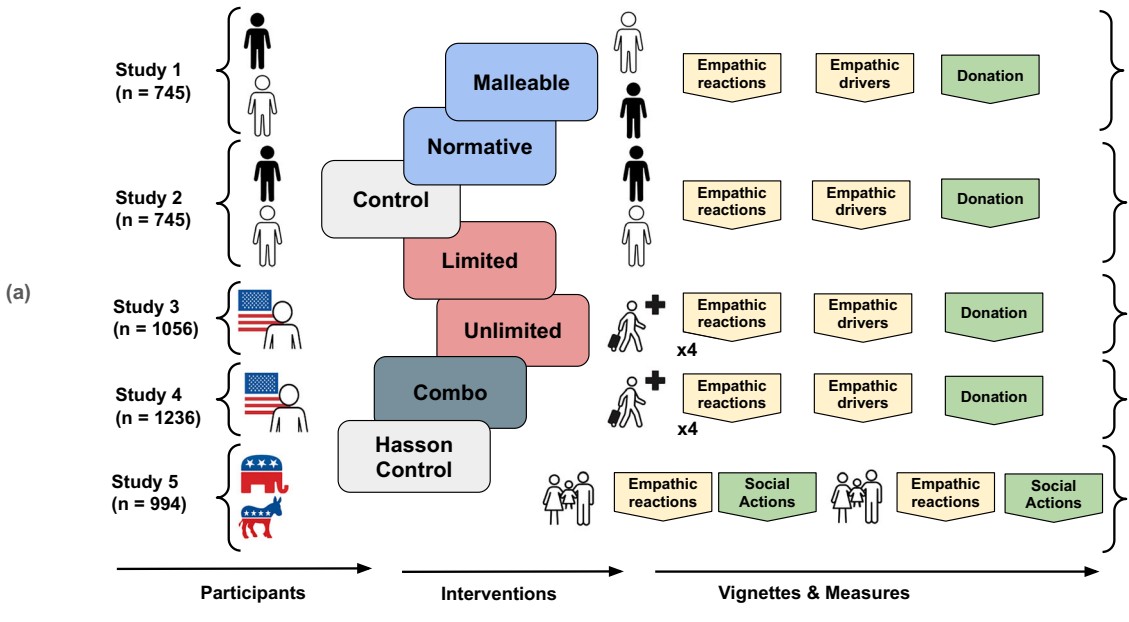

(a)

(b)

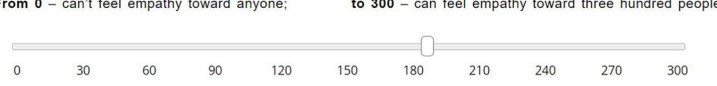

**Definition:** Empathy is defined as the ability to understand and share the feelings and thoughts of others. For example, empathizing with someone in distress involves understanding the situation from his/her perspective and feeling his/her negative emotions.

Recent studies have found that *empathy is an unlimited resource* so people *can feel* it toward a large number of people.

Imagine that you are about to meet people in distress. Toward how many of them could you feel empathy?

Please answer how many people you can empathize with by using the scale below:

**From 0** – can't feel empathy toward anyone;     **to 300** – can feel empathy toward three hundred people.

0    30    60    90    120    150    180    210    240    270    300

**Fig. 1 | showing overview and design of studies and the intervention Unlimited a** In Study 1, Black and white American participants were randomly assigned to one of five conditions. Each group was exposed to one intervention or control. Participants then read a vignette about either a Black or a white man suffering from poverty. Participants' and target's ethnicities were mismatched to create an intergroup context. Participants answered the empathic reactions and drivers' questions and decided if they wanted to donate from a bonus payment to help individuals in a situation similar to the vignette. Study 2 was identical to Study 1, but participants' and target's ethnicity were matched, creating an ingroup context. Study 3 assigned American participants into one of five groups as above. After completing the intervention, participants read four randomly presented testimonies from Syrian refugees who had fled to the US. Participants answered empathic reaction questions after each testimony (four ratings in total), and empathic driver questions after the

last testimony. They were then given the opportunity to donate from a bonus payment to help people in similar circumstances as the targets in the testimonies. Study 4 was identical to Study 3, but with less engaging testimonies, and with changes to two of the interventions: Malleable was transformed into a combination of the previous Malleable and Normative interventions and a new Normative intervention was created. In Study 5, we replicated interventions used in Hasson and colleagues' Study 3 (Unlimited, Limited and Hasson Control), as well as the control from Studies 1–4. We used a within-subjects design on Democrat or Republican voters in the latest US presidential election and participants reported single-item empathy, empathic reactions, as well as support for social actions for both in- and out-group members. **b** Screenshot of the intervention Unlimited (studies 1-4), also used by Hasson and colleagues[18].

had improved their empathic accuracy for positive emotions. Furthermore, participants in the malleable condition held stronger beliefs that empathy is malleable compared to control, but with no other downstream effects. Notably, none of the interventions increased empathy for outgroup members. One reason for this might be that these interventions are best understood as targeting participants' approach motives, i.e., motives to further *increase* empathy[1,17,46,47]. Intergroup contexts are instead characterized by 'avoidance motives', i.e., motives to *distance* oneself from empathy[1,17,46,47]. The tendency to empathize less with outgroup targets seems to be mainly driven by antipathy towards them, rather than by extraordinary empathy towards ingroup targets[13,48]. Interventions targeting avoidance motives might therefore be most effective for increasing motivation to empathize in intergroup contexts. To our knowledge, no studies have formally tested how motivated empathy interventions that specifically target approach- and avoidance motives function in the same experimental setting, and the present work addresses this gap.

To effectively compare different interventions targeting different motivations with each other, we used Hasson and colleagues' existing short interventions as well as shortened forms of two interventions from Weisz and colleagues[17]. These were modified to resemble Hasson's previously successful short forms but targeted the motives utilized by Weisz and colleagues: emphasizing that empathy is malleable or that it is desirable and normative. We had several reasons for modelling our study on Hasson and colleagues' brief interventions. First, their unlimited intervention successfully motivated participants to feel more empathy with outgroup members, which is our main domain of interest. Second, given the practical advantages of brief interventions, identifying additional successful brief interventions and comparing their efficacy is important. Third, given the potential impact of empathy interventions that are both successful and speedy, it is imperative to determine to what extent previous findings replicate and generalize to other contexts and groups. Psychology is still coming to terms with the 'replication crisis'[49,50], and strict replication procedures should be implemented also in emerging fields, including motivated empathy interventions. It is similarly important to investigate the generalizability of important findings[51,52]. Although this was partly done by Hasson and colleagues[18] by varying populations and outgroups, we add in- and intergroups from differing ethnicities, i.e., Black and white Americans and Liberal and Democratic American voters.

Over five studies (total *n* = 4776, see Fig. 1a), we examined how effective the short interventions were using participants' self-reports, monetary donations and support for social actions. We hypothesised, in line with recent results[18], that motivated empathy interventions applied in intergroup contexts, would generate more empathy and prosocial behaviour compared to control and empathy-reducing interventions.

## Methods
Our studies comply with all relevant national ethical regulations and are approved by the Swedish ethical review authority: Dnr 2021-05406-01 and Dnr 2023-07964-01. Informed consent was obtained from all participants, and we compensated all participants for their participation. Any donations participants made from their payments were subsequently donated to the relevant organizations (YMCA and IAC).

### Study 1
Study 1 is preregistered at OSF: https://osf.io/js3mx

**Participants**. 745 American participants (338 females, 405 males; 370 Black, 375 white; $M_{age}$ = 41, $SD_{age}$ = 13; sex and ethnicity are self-reported and provided by Prolific), recruited via Prolific for online participation, were included in our analysis. They received 2.11 $ as an average fee for participating. All participants included in the analyses successfully answered at least one of two attention checks. In accordance with our preregistered exclusion criteria, we excluded 140 participants from analysis, as they did not perceive the stimuli as expected (target as belonging to the relevant outgroup). A power calculation showed that we

needed 740 participants to achieve a 95% chance to detect a medium effect size of Cohen's d = 0.42, with α = 0.05 (the same effect size obtained in Hasson and colleagues' Study 2[18]).

**Interventions**. We used four motivated empathy interventions and one control measure, giving us five experimental groups. Two of the interventions had previously been used by Hasson and colleagues' (second and third studies)[18]. These interventions include brief information about empathy being either a limited or an unlimited resource (see Fig. 1b). The two other interventions, Malleable and Normative, were inspired by interventions constructed by Weisz and colleagues'[17] (2021). The intervention Malleable informs participants that empathy is malleable, and that people can become more empathic if they try[53]. Normative, instead, informs participants that empathy is socially desirable and that people in most communities value empathy and expect members to be empathic. Finally, the control intervention states facts about financial investments and does not mention empathy. All interventions ask a follow-up question answered with a visual analogue scale (see Table SR1 in Supplementary results for a full description of the interventions).

The number of participants differed slightly between experimental conditions due to participants failing attention checks, being excluded from analysis because they did not perceive the target as expected, i.e., as an outgroup member, or randomized allocation of participants between conditions: Normative = 151; Unlimited = 144; Limited = 149; Malleable = 139; Control = 162.

**Stimuli**. We used two versions of a vignette depicting an American man suffering from poverty as experimental stimuli. The vignette briefly describes the background of a man visiting the YMCA for help with basic needs. In one version the vignette mentioned stereotypical Black American names and placed the man in Philadelphia. The other mentions stereotypical white American names and Boston as a setting. In a pre-test the two versions of the vignette were similarly rated on how affected and motivated to help participants were by the vignette, the vignette's relatability and readability, and how realistic it was perceived to be (see https://osf.io/hr6gn/ for stimuli validation data).

**Outcome measures**. Participants self-reported their empathy. The compound measure Empathic reactions consisted of three items, asking participants "how much sympathy/empathy/compassion did you feel with X"? Hasson and colleagues[18] use both the empathic reactions measure and, occasionally, a single-item empathy question, so we used both in our subsequent analyses. The compound measure Empathic drivers was constructed to capture factors that are often predictive of empathy (such as affect, relatability, and motivation to help[2]). It also consisted of three items: "how badly affected were you by the story about X"; "how much did you relate to X as you read his story"; "how motivated are you to help X improve his situation if you are given the opportunity to do so?" All items were rated with seven-point Likert scales and measures averaged across items. In addition, prosocial behaviour was measured by offering participants to use their 1 £ bonus payment, to make a monetary donation to YMCA (featuring in the vignette). Donations were made on a visual analogue scale, with ten-pence intervals and a range of 0-100 pence (see Supplementary Table SR1.1 for full description of donation measures).

**Procedure**. Using a between-subjects design we randomly assigned participants to one of the five experimental conditions. Participants were paired with a target that stereotypically did not match their stated ethnicity on Prolific. Participants' that nevertheless rated the target as matching their own ethnicity were removed from the analysis, to ensure that they perceived an intergroup context. Participants were informed that they would participate in an experiment about poverty and were asked to read all texts thoroughly. In each experimental condition, participants were exposed to the relevant motivated empathy intervention

(or control). After an attention check, participants read the vignette. They subsequently answered empathic reactions, including single-item empathy, and empathic drivers, and were given the opportunity to donate to the YMCA. Participants then indicated what ethnicity they perceived the target to have and answered questions regarding the degree of their beliefs in different aspects of empathy, related to the interventions (all manipulation checks are reported in Supplementary Table SR1.3). Subsequently, participants rated their trust in the YMCA. At the end of the experiment, participants were given more information about its purpose.

**Analysis.** The outcome measures empathic reactions and empathic drivers both showed a high internal consistency (empathic reactions: Cronbach's α = 0.93, empathic drivers: Cronbach's α = 0.81), and were included in the analysis. All analyses were performed in the *R* statistical language (version 4.3.1) using the *brms* package[54]. Following our pre-registration, we used a Bayesian multiple regression model with a Gaussian link function to analyse the effect of interventions on the manipulation check, empathy measures and donations. For each model, interventions were analysed as an independent categorical variable with five levels and manipulation check/empathic measure/donation was used as the dependent variable. Dependent variables were z-scored prior to analysis. All models used standard normal priors (*normal(0,1)*) for all levels of the independent variable, suppressing the global intercept, and an *exponential(1)* prior for the residual variation (*sigma*).

We computed contrasts between interventions for all measures and report (sometimes in the Supplementary results) computed contrasts and their 95% Highest Density Interval (HDI). We will conclude the presence of a hypothesized difference if the resulting 95% highest density interval (HDI) excludes 0. To quantify relative evidence of an effect for reported contrasts, we computed Bayes factors, calculated as the Savage-Dickey ratio evaluated at 0 (null). We interpret Bayes Factors > =3 as weak evidence for an effect and > =10 as strong evidence for an effect; Bayes Factors < = 0.33 as weak evidence for null and < = 0.10 as strong evidence for the null, following standard recommendations[55]. Data distribution was assumed to be normal, but this was not formally tested.

## Study 2
Study 2 is preregistered at OSF: https://osf.io/kpesd.

**Participants.** 745 American participants (359 females, 384 males; 368 Black, 377 white; $M_{age}$ = 39, $SD_{age}$ = 13; sex and ethnicity are self-reported and provided by Prolific), recruited via Prolific for online participation, were included in our analysis. They received 2.18 $ as an average fee for participating. All participants included in the analyses successfully answered at least one of two attention checks. In accordance with our preregistered exclusion criteria, we excluded 140 participants from analysis, as they did not perceive the stimuli as expected (target as belonging to the relevant ingroup). A power calculation showed that we needed 740 participants to achieve a 95% detecting a medium effect size of Cohen's d = 0.42, with α = 0.05 (same effect size obtained in Hasson and colleagues' Study 2[18]). The number of participants differed slightly between experimental conditions due to participants failing attention checks, being excluded from analysis because they did not perceive the target as expected, i.e., as an ingroup member, or randomized allocation of participants between conditions: Normative = 157; Unlimited = 154; Limited = 142; Malleable = 146; Control = 146.

**Outcome measures and procedure.** We used the same materials and measures as in Study 1. Similarly, the design and procedure were identical to Study 1, except for one important difference: based on their stated ethnicity on Prolific, participants were matched with a pre-supposed *matching* version of the target in the vignette to create an ingroup context.

**Analysis.** The outcome measures empathic reactions (Cronbach's α = 0.92) and empathic drivers (Cronbach's α = 0.8) showed a high internal consistency and were both included in the analysis. As per our preregistration, all models were the same as in Study 1. We calculated paired contrasts with associated Bayes Factors and interpreted evidence as in Study 1. Data distribution was assumed to be normal, but this was not formally tested.

## Study 3
Study 3 is preregistered at OSF: https://osf.io/f5zds.

**Participants.** 1056 American participants (638 females, 412 males; $M_{age}$ = 38, $SD_{age}$ = 13; sex is self-reported and provided by Prolific; ethnicity is part of the standard demography data provided by Prolific but not of interest in this study and therefore nor analysed), were recruited via Prolific for online participation, and included in the analysis. They received 2.66 $ as an average fee for participating. Participants included in the analyses successfully answered at least one of two attention checks. A power calculation showed that we needed 1050 participants to achieve a 95% chance of detecting a medium effect size of Cohen's d = 0.22, with α = 0.05 (the same effect size obtained in our first study, which is approximately half the original effect size found by Hasson and colleagues in their second study).

The number of participants differed slightly between experimental conditions due to participants failing attention checks or randomized allocation of participants between conditions: Normative = 221; Unlimited = 206; Limited = 208; Malleable = 210; Control = 221.

**Stimuli and outcome measures.** In this study, we used stimuli from Hasson and colleagues (second study)[18]. It consists of four testimonies that are based on real stories from Syrian refugees that had fled to the US. Testimonies were presented in a randomized order. We used the same interventions and measures as in Studies 1 and 2, except for changing the organization used for donations to the IAC, an organization that helps migrants in the US with basic needs.

**Procedure.** The procedure was similar to Study 1. Participants completed their intervention, answered an attention-check, and read four testimonies from Syrian refugees. After each testimony, participants rated their empathic reactions, including single-item empathy (resulting in a total of four ratings). Participants rated empathic drivers for the last testimony in the trial only. They decided whether they wanted to donate to help individuals in a similar situation as the target, this time with donations directed to the IAC. They answered a second attention-check, rated their beliefs about empathy (see Supplementary Table SR1.3 for manipulation checks), and their trust in IAC. Finally, participants were given more information about the purpose of the experiment.

**Analysis.** Following our preregistered analyses, we reused the models from Study 1 for manipulation checks, empathic drivers, and donations. Both empathic reactions (Cronbach's α = 0.95) and empathic drivers (Cronbach's α = 0.74) were included in the analysis. The measures single-item empathy and empathic reactions were repeated in this study and therefore we used a Bayesian multilevel model with a Gaussian link function for the analysis, as preregistered. For population-level effects, empathy/empathic reactions were used as the dependent variable and z-scored. Interventions were analysed as a categorical independent variable with five levels. We used participants and stimuli as grouping effects, each with a global intercept. For priors, we used standard normal priors (*normal(0,1)*) for all levels of the independent variable, suppressing the global intercept, and an *exponential(1)* prior for the residual variation (*sigma*). For each grouping effect, we used standard normal priors (*normal(0,1)* and used an exponential prior *(1)* for the residual variation (*sigma*). This model differs from the slightly more complex model described in our preregistration, as that model did not converge. We

calculated paired contrasts with associated Bayes Factors and interpreted evidence as in Study 1. Data distribution was assumed to be normal, but this was not formally tested.

In Study 3, we mainly focused on replicating work by Hasson and colleagues, now using their original stimulus from their second study. We erroneously excluded manipulation checks for the interventions Normative and Malleable. Due to potential ceiling effects, we conducted a second data collection with slightly attenuated stimuli in Study 4 and in this study, we did include manipulation checks for all interventions.

## Study 4
Study 4 is preregistered at: https://osf.io/k4dvc.

**Participants.** 1236 American participants (621 females, 615 males; $M_{age} = 36.9$, $SD_{age} = 13$; sex is self-reported and provided by Prolific. Ethnicity is part of the standard demography data provided by Prolific but not of interest in this study and therefore nor analysed), were recruited on Prolific for online participation, and included in the analysis. They received 2.60 $ as an average fee for participating. Participants included in the analyses successfully answered at least one of two attention checks. A power calculation showed that we obtained above 95% chance of detecting a medium effect size of Cohen's $d = 0.22$, with $\alpha = 0.05$ by allocating 240 participants per condition.

The number of participants differed slightly between experimental conditions due to participants failing attention checks or randomized allocation of participants between conditions: Normative = 239; Unlimited = 246; Limited = 242; Combo = 257; Control = 252.

**Interventions.** The interventions Unlimited, Limited and Control were kept as in studies 1-3, but we modified the content of Normative to increase the emphasis on the normativity of empathy and combined the previous Normative and Malleable interventions into a combined intervention (Combo; see Supplementary Table SR1 for a full description of the interventions).

**Stimuli and outcome measures.** To avoid potential ceiling effects, we adjusted the stimuli used in Study 3. The testimonies were altered so that targets did not primarily report having experienced difficulties themselves, but to have observed others do so. In addition, their status was changed from refugees to legal migrants.

We used the same empathic drivers and donations measures as in Study 3. For empathic reactions we changed the formulation slightly and asked participants about their *current* empathy with the legal migrants ("Following the story, to what extent do you currently feel empathy toward X?" From 1 – Not at all; to 7 – Very much). We also added new manipulation checks (see Supplementary Table SR1.3).

**Procedure.** The procedure was like the one in Study 3, but with extra manipulation checks (e.g. ratings of beliefs about empathy) at the end of the experiment.

**Analysis.** Both empathic reactions (Cronbach's $\alpha = 0.92$) and Empathic drivers (Cronbach's $\alpha = 0.82$) were included in the analysis.

We used the same models as in Study 3, but with adjustments to the priors of the population-level effects of all models, as can be seen in our preregistration. We used the appropriate dependent variable (empathy/empathic reactions) and condition as a five-level categorical variable. Grouping factors ID codes for participant ID and name codes for stimulus (story) id. For the independent variable, we set a prior (normal (0, 0.2) and for sigma (exponential (1). For grouping factors, we set priors to (exponential (1) and standard deviations for grouping factors (exponential 10).

We calculated paired contrasts with associated Bayes Factors and interpreted evidence as previously. Data distribution was assumed to be normal, but this was not formally tested.

## Study 5
Study 5 is preregistered at: https://osf.io/vu9td.

**Participants.** 994 American participants (510 females, 483 males; $M_{age} = 40.1$, $SD_{age} = 13$), that either voted Democrat (498) or Republican (496) in the 2024 presidential election, were recruited via Prolific (sex and voting behaviour are self-reported and provided via Prolific. Ethnicity is part of the standard demography data provided by Prolific but not of interest in this study and therefore nor analysed) for online participation, and included in the analysis. They received 1.96 $ as a fee for participating. Participants included in the analyses successfully answered at least one of two attention checks. A power calculation gave us a power of ≥99% to detect the effect size for the target Identity x Condition interaction for empathic reactions found in Hasson and colleagues' Study 3 (Cohen's $d = 0.5$). The number of participants differed slightly between experimental conditions due to participants failing attention checks or randomized allocation of participants between conditions: Unlimited = 249; Limited = 246; Hasson Control = 250; Control = 249.

**Interventions.** The interventions Unlimited, Limited and Hasson Control were replicated from Hasson and colleagues' Study 3. The Control condition was kept the same as in studies 1–4 (see Supplementary Table SR1 for a full description of the interventions).

**Stimuli and outcome measures.** We re-used the stimuli from Hasson and colleagues' Study 3, with adjustments to fit a US context, e.g., names of the people in need and the cities they lived in. To create an in/outgroup context, we added that they voted for either the Democrats or the Republicans in the last US election instead of the Israeli/Arab context used by Hasson. We also changed the families' pictures. As in the original study, the identity of the family, the difficulty they were facing (health or financial) and the order of presentation were all counterbalanced. Both Democrats and Republicans were used as in- and outgroup targets.

We used the same empathic reactions measure as Hasson and colleagues' Study 3. We also used the same prosocial behaviour measure, with some slight adjustments to fit a US context (Following the article, to what extent do you currently agree with the following statements below: "Support the authorities in providing help to this family"; "Promoting policies that improve the condition of families in a similar situation"; "The family should get assistance from volunteer organizations"; "The relevant congressperson to handle such cases should be contacted") and all statements were answered using a seven-point Likert scale: from 1 = not at all, to 7 = very much.

**Procedure.** Using a within-participant design, we divided participants into four intervention conditions: Unlimited, Limited, Hasson Control, and Control. After going through their intervention, participants completed an attention check before reading about a struggling American family that voted for Democrats/Republicans in the last election. Following the vignette, participants rated their empathic reactions and support of prosocial actions towards the family. They then read a second vignette about another family with the opposite political affiliation and rated their empathic reactions and support for prosocial action towards them. After reading about the second family, participants completed a second attention check and then a manipulation check. They finished the experiment by filling out the Interpersonal Reactivity Index[56,57] before being briefed about the experiment and its purpose.

**Analysis.** Both empathic reactions (Cronbach's $\alpha = 0.92$) and Social Action (Cronbach's $\alpha = 0.87$) were included in the analysis.

We used similar models to the ones used in Study 4. For models used to calculate contrasts across contexts, we we added an interaction between conditions and a two-level variable, stimulus, denoting in- and outgroup targets. We used the appropriate dependent variable (empathy/empathic reactions/social action). Grouping factors ID codes for participant ID and Stimuli codes for stimulus ID. For the independent variable, we set a prior

*(normal (0, 0.2)* and for *sigma (exponential (1).* For priors on the standard deviation of effects varying by participant, we set priors to *exponential (1)* and for effects varying by stimulus, we set priors to *exponential (10).*

We calculated paired contrasts with associated Bayes Factors and interpreted evidence as in previous studies. Data distribution was assumed to be normal, but this was not formally tested.

The IRI scores were summarized for each participant per sub-scale: personal distress, empathic concern, perspective taking and fantasy. Questions: 3,4,7,12,13,14,15,18,19 were reverse scored.

### Reporting summary
Further information on research design is available in the Nature Portfolio Reporting Summary linked to this article.

## Results
### Study 1
In Study 1 we tested how our interventions would affect empathy in an intergroup context. To do so we recruited 745 (370 Black, 375 white; ethnicity is self-reported and provided by Prolific) participants via Prolific, giving us 95% power to find the same effect size (Cohen's d = 0.42) as reported in Hasson and colleagues' study 2. As Hasson and colleagues, we focused on intergroup empathy and a context consisting of different ethnicities, i.e., Black and white American participants. The aim was to generalize findings regarding previously successful empathy interventions. Ethnicity is known to produce intergroup contexts[58,59], and to affect empathy in these contexts[26,60,61]. See Table 1 for descriptive statistics.

Given the results from previous studies using motivated empathy interventions[17,18], we preregistered hypotheses that the Unlimited and Malleable interventions would generate the most empathy (reactions and drivers) and prosocial behaviour, followed by the intervention Normative, and then Limited and Control. To test our hypotheses, we contrasted the interventions and control against each other to calculate differences in their associations with empathy and donations. When the 95% highest density interval (HDI) excludes zero, we interpret that as support for an effect, and we report Bayes factors to quantify the relative evidence for such an effect.

**Effects of beliefs about empathy being (un)limited.** We first assessed the efficacy of the interventions based on Hasson et al.[18], by regressing manipulation check scores against condition and then computing paired contrasts between the posterior estimates of each condition. Participants in the Unlimited condition had stronger beliefs that empathy is not a limited resource than participants in the Control and Limited conditions. Participants in Control also had stronger beliefs about empathy not being a limited resource than participants in Limited (see Supplementary Table SR1.3).

For our analyses, we fitted Bayesian regression models, using the relevant empathy/donation measure as outcome, including single-item empathy, as that was used by Hasson and colleagues[18], see Methods for model specification. We used the model coefficients to compute contrasts between conditions for the four outcome measures (see Fig. 2). We did not find any difference between Unlimited and Control for single-item empathy ($b_{diff} = -0.02$, $SE = 0.11$, $HDI = [-0.25$ to $0.20]$, $BF_{10} = 0.08$), empathic reactions ($b_{diff} = -0.04$, $SE = 0.11$, $HDI = [-0.26$ to $0.19]$, $BF_{10} = 0.08$), empathic drivers ($b_{diff} = 0.07$, $SE = 0.11$, $HDI = [-0.16$ to $0.29]$, $BF_{10} = 0.10$), or, donation ($b_{diff} = -0.02$, $SE = 0.12$, $HDI = [-0.25$ to $0.20]$, $BF_{10} = 0.08$). On the other hand, the Limited condition reduced both single-item empathy ($b_{diff} = -0.28$, $SE = 0.11$, $HDI = [-0.5$ to $-0.06]$, $BF_{10} = 1.76$) and empathic reactions compared to Control ($b_{diff} = 0.25$, $SE = 0.11$, $HDI = [-0.47$ to $-0.03]$, $BF_{10} = 0.99$), but not empathic drivers ($b_{diff} = -0.07$, $SE = 0.11$, $HDI = [-0.30$ to $0.15]$, $BF_{10} = 0.10$), or, donations ($b_{diff} = -0.01$, $SE = 0.11$, $HDI = [-024$ to $0.21]$, $BF_{10} = 0.08$). Additionally, we found a difference for single-item empathy ($b_{diff} = 0.26$, $SE = 0.12$, $HDI = [0.04–0.49]$, $BF_{10} = 1$) when contrasting Unlimited against Limited, but not for empathic reactions ($b_{diff} = 0.22$, $SE = 0.12$, $HDI = [-0.01$ to $0.45]$, $BF_{10} = 0.44$), empathic drivers ($b_{diff} = 0.14$, $SE = 0.12$, $HDI = [-0.09$ to $0.37]$,

$BF_{10} = 0.17$), or, donation ($b_{diff} = -0.01$, $SE = 0.12$, $HDI = [-0.24$ to $0.22]$, $BF_{10} = 0.08$).

**Effects of inducing motivational beliefs about empathy.** We continued to assess the two additional interventions' manipulation checks, targeting that empathy is malleable or normatively desirable. The contrast between Normative and Control regarding belief in empathy's normative desirability did not exclude zero in the Highest Density Interval, and the same was true for the contrast between Malleable and Control regarding belief in its malleability (See Supplemental SR 2.5 and SR 2.7).

Malleable and Normative did not induce appropriate beliefs above Control, we nevertheless contrasted them against the Control condition on the outcome measures (Fig. 2). Neither intervention was successful. For the Normative condition, there were no differences in single-item empathy ($b_{diff} = -0.04$, $SE = 0.11$, $HDI = [-0.25$ to $0.18]$, $BF_{10} = 0.08$), empathic reactions ($b_{diff} = -0.07$, $SE = 0.11$, $HDI = [-0.29$ to $0.15]$, $BF_{10} = 0.10$), empathic drivers ($b_{diff} = 0.16$, $SE = 0.11$, $HDI = [-0.07 – 0.38]$, $BF_{10} = 0.20$), or, donations ($b_{diff} = 0.15$, $SE = 0.11$, $HDI = [-0.07$ to $0.37]$, $BF_{10} = 0.19$) compared to Control. Similarly, for the contrast between Malleable and Control conditions, we found no differences in single-item empathy ($b_{diff} = -0.11$, $SE = 0.12$, $HDI = [-0.34$ to $0.12]$, $BF_{10} = 0.12$), empathic reactions ($b_{diff} = -014.$, $SE = 0.12$, $HDI = [-0.37$ to $0.08]$, $BF_{10} = 0.18$), empathic drivers ($b_{diff} = 0.03$, $SE = 0.12$, $HDI = [-0.20$ to $0.26]$, $BF_{10} = 0.08$), or, donations ($b_{diff} = 0.00$, $SE = 0.12$, $HDI = [-0.23$ to $0.23]$, $BF_{10} = 0.08$).

As none of our interventions outperformed Control for empathy or donations, we no longer had an interest in examining interactions between these variables, as was preregistered. We also used the measure "Empathic beliefs" as single-item manipulation checks, which is a deviation from our pre-registration.

**Comparing approach and avoidance motives.** To understand how interventions targeting approach and avoidance motives compare in an intergroup context we compared the avoidance-reducing Unlimited and the approach-increasing Malleable and Normative with each other. No contrasts excluded zero in the Highest Density Interval for Normative and Unlimited on any of the measures single-item empathy, empathic reactions, empathic drivers, or donations. The same held for the contrast between Malleable and Unlimited (all contrasts are reported in Supplemental Study 1, Tables SR2.9, SR2.11, SR2.13, and SR2.15).

**Conclusion Study 1.** In sum, none of the interventions appear to have generated more empathy towards outgroup targets than an unrelated control intervention. No differences were found for empathy measured through self-reports (single-item empathy, empathic reactions and empathic drivers), or for prosocial behaviour towards outgroup targets, measured as monetary donations. Additionally, approach-increasing and avoidance-decreasing interventions performed similarly in an intergroup context. Thus, we do not confirm our hypotheses that interventions generate more empathy (reactions and drivers) and prosocial behaviour than Control.

Even if increasing empathy towards outgroup targets is our primary concern in this work, it is possible that the results in Study 1 depended on the intergroup context used, so we next replicated the study using an ingroup context instead.

### Study 2
In our second study, we recruited 745 (368 Black, 377 white; ethnicity is self-reported and provided by Prolific) participants via Prolific to test the four motivated empathy interventions and control intervention in an ingroup context. This gave us 95% power to find a medium effect size (Cohen's d = 0.42). People are inclined to feel empathy with persons they perceive as ingroup members[1], and interventions targeting approach motives might be best suited for an ingroup context. Specifically, interventions were inspired by Weisz and colleagues' work, but differed substantially from their

**Table 1 | Title: Descriptive statistics for studies 1-5**

| STUDY | CONDITION | EMPATHY | EMPATHIC REACTIONS | EMPATHIC DRIVERS | DONATIONS | PROSOCIAL ACTION | MALLEABILITY | MOTIVATION | EMPATHY UNLIMITED | NORMATIVITY |
|---|---|---|---|---|---|---|---|---|---|---|
| 1 | Unlimited | 5.47 (1.54) | 5.51 (1.44) | 2.93 (1.07) | 39.8 (37.5) | Na | 5.06 (1.62) | 4.99 (1.76) | 5.33 (1.96) | Na |
|   | Limited | 5.07 (1.55) | 5.2 (1.43) | 2.78 (1.03) | 40.2 (38.3) | Na | 5.03 (1.67) | 4.76 (1.72) | 4.31 (2.01) | Na |
|   | Normative | 5.44 (1.45) | 5.46 (1.34) | 3.02 (0.99) | 46.5 (38.6) | Na | 5.13(1.52) | 4.93 (1.66) | 4.74 (2.2) | Na |
|   | Malleable | 5.34 (1.47) | 5.35 (1.42) | 2.89 (1.1) | 40.6 (38.6) | Na | 5.25 (1.47) | 4.87 (1.81) | 4.7 (2.18) | Na |
|   | Control | 5.5 (1.52) | 5.56 (1.45) | 2.86 (1.09) | 40.7 (40.4) | Na | 4.98 (1.64) | 4.64 (1.87) | 4.83 (2.25) | Na |
| 2 | Unlimited | 5.45 (1.44) | 5.48 (1.33) | 2.86 (1.07) | 48.5 (39.8) | Na | 4.94 (1.65) | 4.85 (1.74) | 5.36 (1.86) | Na |
|   | Limited | 5.32 (1.5) | 5.43 (1.35) | 2.91 (0.96) | 39.6 (38.4) | Na | 5.01 (1.6) | 4.97 (1.75) | 4.63 (1.94) | Na |
|   | Normative | 5.57 (1.33) | 5.63 (1.23) | 2.97 (1.02) | 48.3 (38.5) | Na | 5.29 (1.44) | 5.19 (1.63) | 4.92 (2.11) | Na |
|   | Malleable | 5.46 (1.43) | 5.49 (1.35) | 2.88 (1.07) | 44.5 (36) | Na | 5.44 (1.32) | 5.04 (1.6) | 5.22 (1.92) | Na |
|   | Control | 5.64 (1.25) | 5.72 (1.19) | 2.94 (0.9) | 49.7 (37.6) | Na | 5.2 (1.52) | 5.10 (1.53) | 5.16 (1.83) | Na |
| 3 | Unlimited | 6.16 (1.31) | 6.2 (1.2) | 4.49 (1.49) | 51.6 (41.7) | Na | Na | Na | 5.54 (1.84) | Na |
|   | Limited | 5.85 (1.41) | 6.03 (1.15) | 4.57 (1.4) | 54.7 (42) | Na | Na | Na | 5.12 (1.82) | Na |
|   | Normative | 6.08 (1.3) | 6.21 (1.16) | 4.68 (1.34) | 57.5 (41.9) | Na | Na | Na | 5.24 (1.85) | Na |
|   | Malleable | 5.91 (1.34) | 6 (1.26) | 4.53 (1.37) | 53.5 (40.4) | Na | Na | Na | 4.95 (1.89) | Na |
|   | Control | 5.93 (1.52) | 6.09 (1.24) | 4.3 (1.52) | 51.5 (41.4) | Na | Na | Na | 5.11 (1.96) | Na |
| 4 | Unlimited | 4.94 (1.58) | 4.96 (1.52) | 3.66 (1.56) | 45.8 (40.24) | Na | 5.08 (1.52) | Na | 4.74 (1.53) | 5.93 (1.21) |
|   | Limited | 4.9 (1.64) | 4.92 (1.56) | 3.76 (1.48) | 54.15 (39.64) | Na | 5.11 (1.43) | Na | 3.92 (1.56) | 5.75 (1.37) |
|   | Normative | 5.11 (1.51) | 5.12 (1.46) | 3.87 (1.51) | 51.95 (40) | Na | 5.23 (1.26) | Na | 3.99 (1.44) | 5.81 (1.18) |
|   | Combo | 4.88 (1.5) | 4.93 (1.44) | 3.76 (1.43) | 47.04 (39.05) | Na | 5.31 (1.23) | Na | 3.85 (1.37) | 5.75 (1.13) |
|   | Control | 5.14 (1.57) | 5.22 (1.45) | 3.85 (1.45) | 46.73 (41.31) | Na | 5.15(1.41) | Na | 3.96 (1.47) | 5.76 (1.24) |
| 5 | Unlimited | 5.6 (1.43) | 5.52 (1.39) | Na | Na | 5.33 (1.46) | Na | Na | 4.95 (1.52) | Na |
|   | Limited | 5.28 (1.59) | 5.31 (1.47) | Na | Na | 5.22 (1.44) | Na | Na | 3.46 (1.56) | Na |
|   | Hasson Control | 5.68 (1.37) | 5.67 (1.34) | Na | Na | 5.47 (1.3) | Na | Na | 4.39 (1.5) | Na |
|   | Control | 5.51 (1.53) | 5.52 (1.46) | Na | Na | 5.35 (1.42) | Na | Na | 4.38 (1.46) | Na |

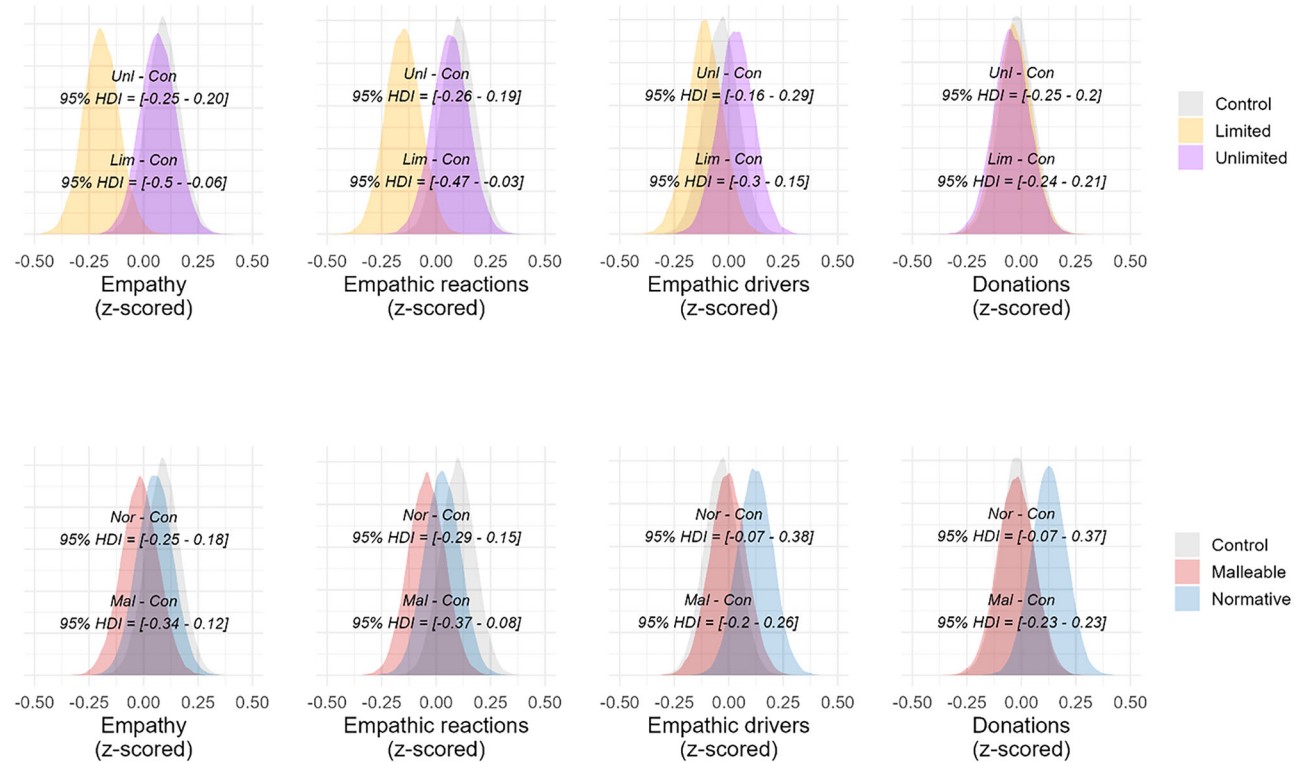

**Fig. 2 | Density distributions of empathy measures and donations from Study 1.** Density distributions (all z-scored) from multiple regression models for single-item empathy, empathic reactions, empathic drivers, and donations in Study 1. In the first row, the bottom HDI interval shows the contrast between Limited and Control, while the top HDI interval shows the contrast between Unlimited and Control. In the second row, the bottom HDI shows the contrast between Malleable and Control, and the top HDI shows the contrast between Normative and Control. Figure based on participants from each condition: Normative = 151; Unlimited = 144; Limited = 149; Malleable = 139; Control = 162.

interventions with respect to length and type of activities[17]. We used the same statistical approach and procedure as in Study 1, with one exception: we matched participants' stated ethnicity with the protagonist's stereotyped ethnicity and only included participants who perceived the protagonist as belonging to the same racial group as themselves in the analysis. See Table 1 for descriptive statistics.

For study 2, we preregistered that the interventions would produce more empathy (reactions and drivers) and prosocial behaviour than Control but not differ from each other.

**Effects of beliefs about empathy being (un)limited.** We began by examining how the Unlimited or Limited interventions affected participants' beliefs about empathy being a limited resource. Participants believed more strongly in the unlimited nature of empathy in the Unlimited and Control conditions than they did in Limited. The contrast between Unlimited and Control did not exclude zero in the Highest Density Interval (see Supplemental Study 2).

We used the same Bayesian models as in Study 1 for our analyses. To assess the effects of the Unlimited and Limited conditions on our outcome measures, we contrasted them against the Control condition (see Fig. 3). We found no differences in the contrast between Unlimited and Control for single-item empathy ($b_{diff} = -0.13$, $SE = 0.11$, $HDI = [-0.36$ to $0.09]$, $BF_{10} = 0.16$), empathic reactions ($b_{diff} = -0.18$, $SE = 0.12$, $HDI = [-0.41$ to $0.05]$, $BF_{10} = 0.27$), empathic drivers ($b_{diff} = -0.08$, $SE = 0.12$, $HDI = [-0.31$ to $0.15]$, $BF_{10} = 0.10$), or, donations ($b_{diff} = -0.03$, $SE = 0.12$, $HDI = [-0.26$ to $0.20]$, $BF_{10} = 0.08$). Similarly, we found no differences between the Limited and Control conditions for single-item empathy ($b_{diff} = -0.23$, $SE = 0.12$, $HDI = [-0.46$ to $0.00]$, $BF_{10} = 0.61$), empathic reactions ($b_{diff} = -0.22$, $SE = 0.12$, $HDI = [-0.45$ to $0.01]$, $BF_{10} = 0.50$), or empathic drivers ($b_{diff} = -0.04$, $SE = 0.12$, $HDI = [-0.27$ to $0.19]$, $BF_{10} = 0.08$). We did, however, find a difference in donations: participants in the Limited

condition donated less of their bonus payment compared to participants in the Control condition ($b_{diff} = -0.26$, $SE = 0.12$, $HDI = [-0.03$ to $-0.49]$, $BF_{10} = 0.96$). No differences were found between Unlimited and Limited, that is, for single-item empathy ($b_{diff} = 0.10$, $SE = 0.12$, $HDI = [-0.13$ to $0.33]$, $BF_{10} = 0.12$), empathic reactions ($b_{diff} = 0.04$, $SE = 0.12$, $HDI = [-0.19$ to $0.27]$, $BF_{10} = 0.09$), empathic drivers ($b_{diff} = -0.04$, $SE = 0.12$, $HDI = [-0.27$ to $0.19]$, $BF_{10} = 0.09$) and donations ($b_{diff} = 0.23$, $SE = 0.12$, $HDI = [0.00-0.46]$, $BF_{10} = 0.52$).

**Effects of inducing motivational beliefs about empathy.** As in Study 1, the contrast between Normative and Control with respect to participants' belief in the normative desirability of empathy did not exclude zero in the Highest Density Interval and the same was true for the contrast between Malleable and Control regarding belief in how malleable empathy is (see Supplemental Study 2).

We nevertheless contrasted Normative and Malleable against the Control condition to assess potential differences they might have on our outcome measures (Fig. 3). For the Normative condition we found no differences in empathy ($b_{diff} = -0.05$, $SE = 0.12$, $HDI = [-0.28$ to $0.17]$, $BF_{10} = 0.09$), empathic reactions ($b_{diff} = -0.06$, $SE = 0.11$, $HDI = [-0.29$ to $0.16]$, $BF_{10} = 0.09$), empathic drivers ($b_{diff} = 0.02$, $SE = 0.12$, $HDI = [-0.20$ to $0.25]$, $BF_{10} = 0.08$), or, donation ($b_{diff} = -0.03$, $SE = 0.12$, $HDI = [-0.26$ to $0.19]$, $BF_{10} = 0.09$) compared to Control. The same pattern was found between Malleable and Control Malleable conditions for single-item empathy ($b_{diff} = -0.13$, $SE = 0.12$, $HDI = [-0.36$ to $0.10]$, $BF_{10} = 0.15$), empathic reactions ($b_{diff} = -0.17$, $SE = 0.12$, $HDI = [-0.40$ to $0.06]$, $BF_{10} = 0.24$), empathic drivers ($b_{diff} = -0.07$, $SE = 0.12$, $HDI = [-0.30$ to $0.16]$, $BF_{10} = 0.10$), or, donation ($b_{diff} = -0.14$, $SE = 0.12$, $HDI = [-0.36$ to $0.09]$, $BF_{10} = 0.16$).

Deviating from our pre-registration, the items in the measure "Empathic beliefs" were once again used as manipulation checks, and we

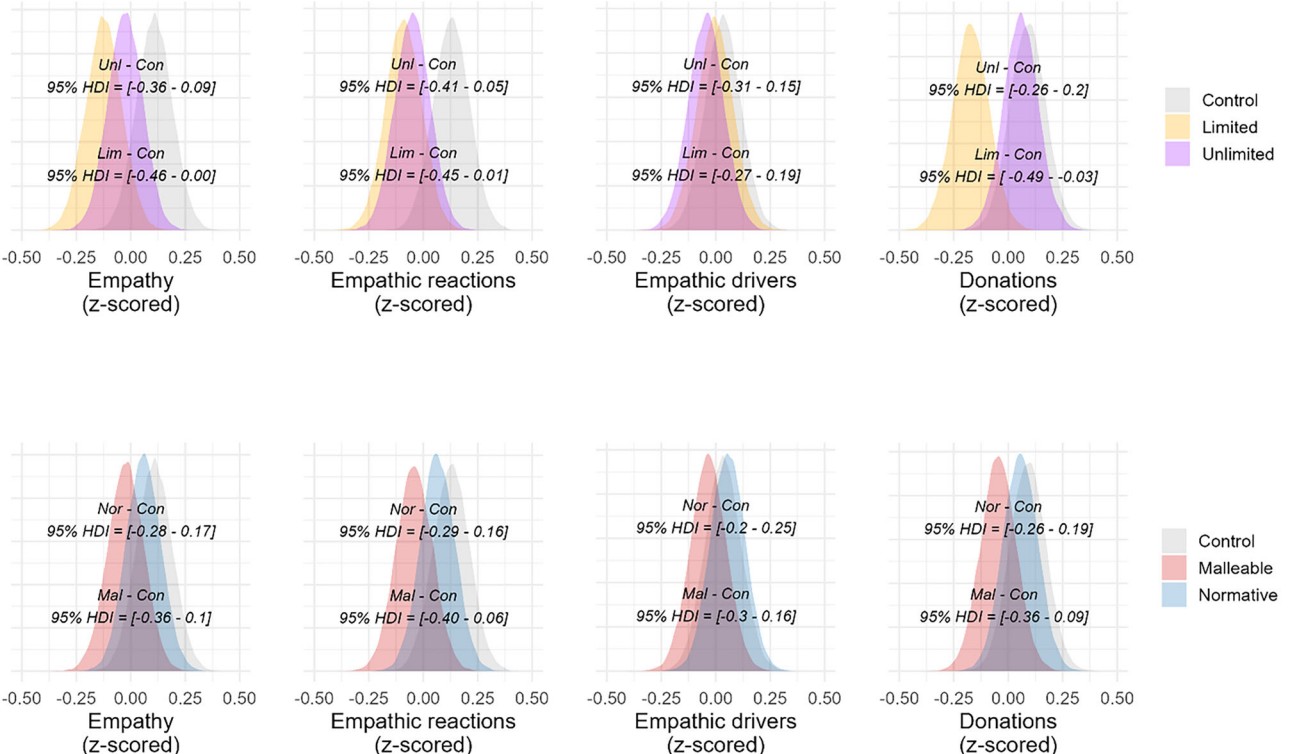

**Fig. 3 | Density distributions of empathy measures and donations from Study 2.** Density distributions (all z-scored) from multiple regression models for single-item empathy, empathic reactions, empathic drivers, and donations in Study 2. In the first row, the bottom HDI interval shows the contrast between Limited and Control, while the top HDI interval shows the contrast between Unlimited and Control. In the second row, the bottom HDI shows the contrast between Malleable and Control, and the top HDI shows the contrast between Normative and Control. Figure based on participants from each condition: Normative = 157; Unlimited = 154; Limited = 142; Malleable = 146; Control = 146.

refrained from further investigating interactions between empathy and donations, as we did not obtain the predicted results.

**Comparing approach and avoidance motives**. We compared the intervention targeting avoidance motives (Unlimited) with the interventions targeting approach motives (Normative and Malleable). This time outcome measures are compared for an ingroup context. As in Study 1, when contrasting Normative against Unlimited it did not exclude zero in the Highest Density Interval for single-item empathy, empathic reactions, empathic drivers or donations. The same was true for the contrast Malleable and Unlimited (all contrasts are reported in Supplemental Study 2, Table SR3.9, SR3.11, SR3.13, and SR3.15).

**Conclusion Study 2**. In sum, none of the interventions generated more empathy towards ingroup targets, regardless of whether it was measured as single-item empathy, empathic reactions, or empathic drivers. Repeating the results from Study 1, no intervention affected prosocial behaviour towards ingroup targets, instead monetary donations were similar to those from participants in the unrelated control intervention. Additionally, we did not find any differences in performance between approach-increasing and avoidance-decreasing interventions in an ingroup context. Thus, we do not confirm our hypotheses that interventions generate more empathy (reactions and drivers) or prosocial behaviour than Control.

**Study 3**

In the first two studies, no intervention was successful when compared to a control condition, regardless of whether they built on increasing approach or decreasing avoidance motives, or whether they were used in an inter- or ingroup context. As previous studies, using the same Limited and Unlimited interventions[18] had shown promising results, this was a surprising finding.

However, as Studies 1 and 2 targeted the generalizability of motivated empathy interventions with novel stimuli and a changed intergroup context, we decided to try to reproduce the effect using the same materials and repeated measures design as in Hasson and colleagues' second study.

We recruited 1056 American participants via Prolific. This gave us a power of 95% to obtain the same effect size as we found in our first study (Cohen's d = 0.22. Note that this is approximately half of the effect size found in Hasson and colleagues' Study 2: Cohen's d = 0.42[18]). We used the same interventions and control conditions as in Studies 1 & 2. See Table 1 for descriptive statistics.

As our first two studies did not show stable differences between interventions, we preregistered that the interventions and Control would generate equal amounts of empathy (reactions and drivers) and prosocial behaviour.

**Effects of beliefs about empathy being (un)limited.** We first examined whether the Unlimited or Limited interventions affected participants' beliefs in the (un)limited nature of empathy, by regressing beliefs about empathy's limits against condition and computing paired contrasts between posterior estimates for each condition. As before, participants in the Unlimited condition reported stronger beliefs that empathy is not a limited resource than participants in Limited and in Control. The contrast between Limited and Control did not exclude zero in the Highest Density Interval (see Supplemental Study 3).

We then conducted the critical test if our study could replicate Hasson and colleagues' results by looking at both single-item empathy and empathic reactions, for which we used a Bayesian multilevel model and used participants and stimuli as grouping factors for varying intercepts, see Methods for model specification. Additionally, we investigated the interventions' effect on empathic drivers and donations using the same models as in Study 1 (See Fig. 4).

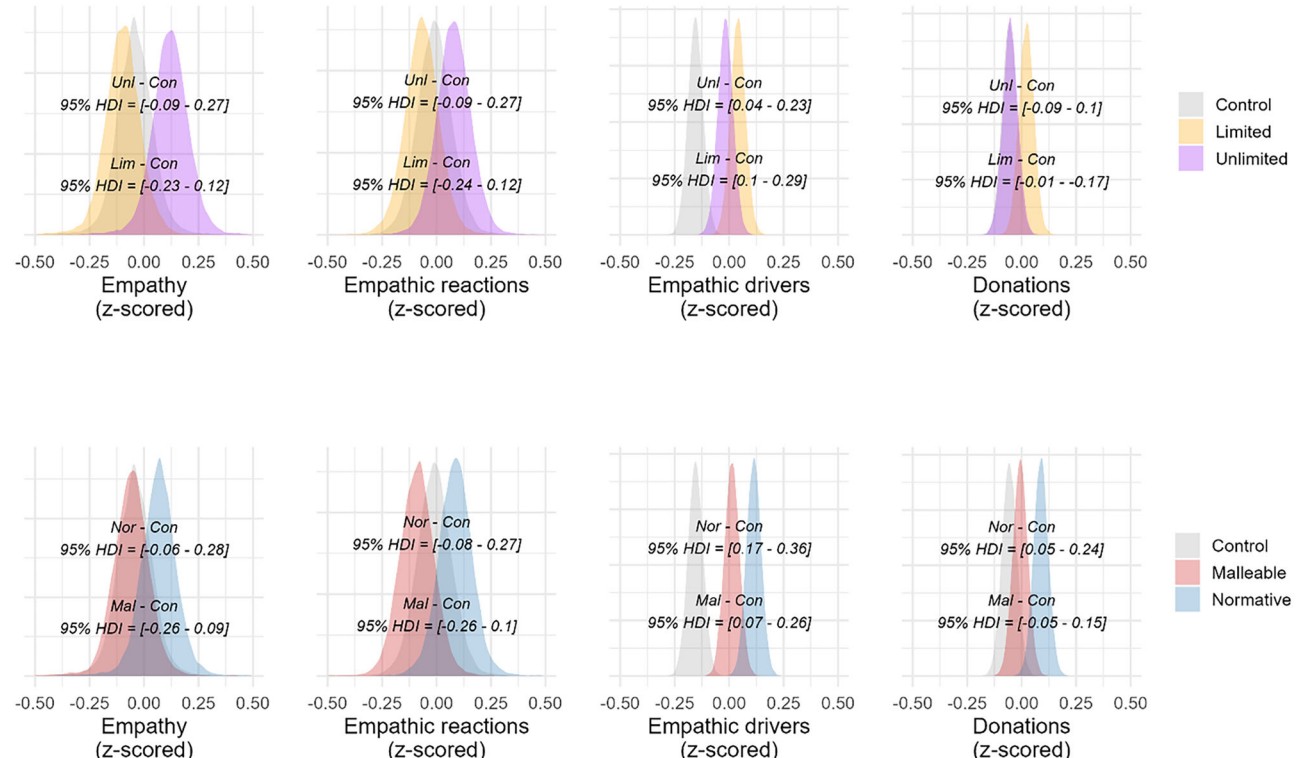

**Fig. 4 | Density distributions of empathy measures and donations from Study 3** Density distributions (all z-scored) from multiple regression models for single-item empathy, empathic reactions, empathic drivers, and donations in Study 3. In the first row, the bottom HDI interval shows the contrast between Limited and Control, while the top HDI interval shows the contrast between Unlimited and Control. In the second row, the bottom HDI shows the contrast between Malleable and Control, and the top HDI shows the contrast between Normative and Control. Figure based on participants from each condition: Normative = 221; Unlimited = 206; Limited = 208; Malleable = 210; Control = 221.

**Effects of beliefs about empathy being (un)limited.** When contrasting Unlimited and Control, we found no differences in single-item empathy ($b_{diff} = 0.09$, $SE = 0.09$, $HDI = [−0.09$ to $0.27]$, $BF_{10} = 0.32$), empathic reactions ($b_{diff} = 0.09$, $SE = 0.09$, $HDI = [−0.09$ to $0.27]$, $BF_{10} = 0.10$) or donations ($b_{diff} = 0.00$, $SE = 0.05$, $HDI = [−0.09$ to $0.10]$, $BF_{10} = 0.04$). However, participants in the Unlimited condition reported stronger empathic drivers ($b_{diff} = 0.14$, $SE = 0.05$, $HDI = [0.04–0.23]$, $BF_{10} = 1.66$) than Control. For Limited against Control, no difference was found for single-item empathy ($b_{diff} = −0.06$, $SE = 0.09$, $HDI = [−0.23$ to $0.12]$, $BF_{10} = 0.08$), empathic reactions ($b_{diff} = -0.06$, $SE = 0.09$, $HDI = [−0.24$ to $0.12]$, $BF_{10} = 0.08$), or donations ($b_{diff} = 0.08$, $SE = 0.05$, $HDI = [−0.01$ tp $0.17]$, $BF_{10} = 0.12$). Surprisingly, also for Limited, participants reported higher empathic drivers than in the Control condition ($b_{diff} = 0.19$, $SE = 0.05$, $HDI = [0.1 – 0.29]$, $BF_{10} = 61.5$). Finally, when contrasting Unlimited and Limited, we found a difference for single-item empathy ($b_{diff} = 0.22$, $SE = 0.09$, $HDI = [0.04 – 0.40]$, $BF_{10} = 1.29$), with participants in the Unlimited condition reporting more empathy. No differences were found for empathic reactions ($b_{diff} = 0.15$, $SE = 0.09$, $HDI = [−0.03$ to $0.33]$, $BF_{10} = 0.23$), empathic drivers ($b_{diff} = -0.06$, $SE = 0.05$, $HDI = [−0.15$ to $0.04]$, $BF_{10} = 0.07$) or donations ($b_{diff} = −0.07$, $SE = 0.05$, $HDI = [−0.17$ to $0.02]$, $BF_{10} = 0.11$).

**Effects of inducing motivational beliefs about empathy.** Due to a coding error, no manipulation checks were included for Normative and Malleable in study 3 (see Method section Study 3).

For this new set of stimuli our Normative and Malleable interventions sometimes differed from Control in how they affected our empathy measures (Fig. 4). First, contrasting the Normative and Control conditions, we did not find a difference for single-item empathy ($b_{diff} = 0.1$, $SE = 0.09$, $HDI = [−0.08$ to $0.27]$, $BF_{10} = 0.13$) or empathic reactions ($b_{diff} = 0.09$, $SE = 0.09$, $HDI = [−0.08$ to $0.27]$, $BF_{10} = 0.11$). On the other hand,

participants in Normative reported higher empathic drivers ($b_{diff} = 0.27$, $SE = 0.05$, $HDI = [0.17–0.36]$, $BF_{10} = > 100$) and made larger donations ($b_{diff} = 0.14$, $SE = 0.05$, $HDI = [0.05–0.24]$, $BF_{10} = 3.72$) than participants in the Control condition. Contrasting Malleable and Control, no differences were found for single-item empathy ($b_{diff} = −0.08$, $SE = 0.09$, $HDI = [−0.26$ to $0.09]$, $BF_{10} = 0.06$), empathic reactions ($b_{diff} = −0.08$, $SE = 0.09$, $HDI = [−0.26$ to $0.1]$, $BF_{10} = 0.1$) or donations ($b_{diff} = 0.05$, $SE = 0.05$, $HDI = [−0.05$ to $0.15]$, $BF_{10} = 0.06$). However, participants in the Malleable condition reported higher empathic drivers than those in Control ($b_{diff} = 0.17$, $SE = 0.05$, $HDI = [0.07–0.26]$, $BF_{10} = 8.29$).

**Comparing approach and avoidance motives.** We also found that the approach-increasing intervention, Normative generated higher empathic drivers and donations when compared with the avoidance-decreasing intervention, Unlimited. No other contrasts excluded zero in the Highest Density Interval (all contrasts are reported in the Supplementary results Study 3, Tables SR4.5, SR4.7, SR4.9 and SR4.11).

**Conclusion Study 3.** In sum, our results suggest that none of the interventions generated more empathic reactions for an outgroup member than control. However, we did find a difference in single-item empathy between Unlimited and Limited, of approximately half of the original effect size (Cohen's d = 0.22, compared to d = 0.42). Importantly, we did not find any difference between the Unlimited condition and Control for single-item empathy, suggesting that the difference between Unlimited and Limited depended on the negative effect Limited had on empathy, rather than a strong positive effect from Unlimited. Thus, we do not confirm our hypotheses, as some interventions generate more empathic drivers and donations than Control.

However, both single-item empathy and empathic reaction scores were generally very high in Study 3, with averages of around 6 on a 7-point scale.

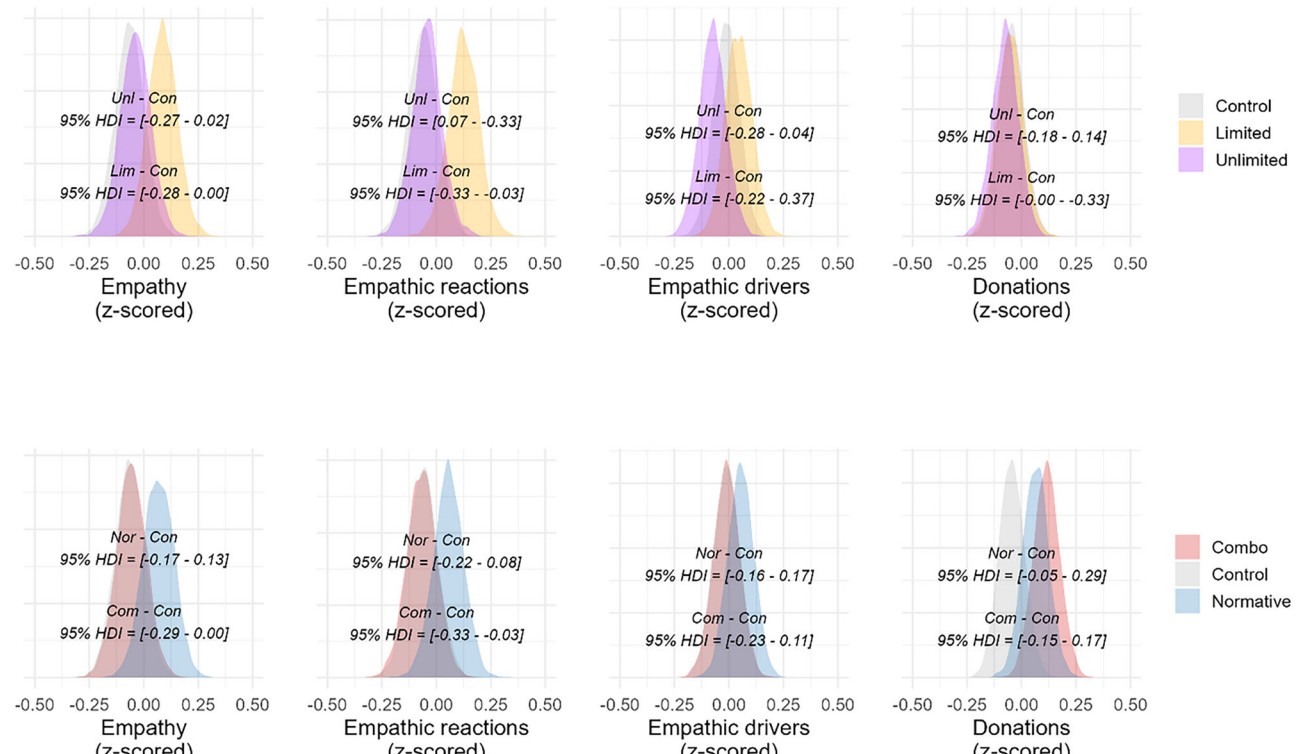

**Fig. 5 | Density distributions of empathy measures and donations from Study 4.** Density distributions (all z-scored) from multiple regression models for single-item empathy, empathic reactions, empathic drivers, and donations in Study 4. In the first row, the bottom HDI interval shows the contrast between Limited and Control, while the top HDI interval shows the contrast between Unlimited and Control. In the second row, the bottom HDI shows the contrast between Combo and Control, and the top HDI shows the contrast between Normative and Control. Figure based on participants from each condition: Normative = 239; Unlimited = 246; Limited = 242; Combo = 257; Control = 252.

There is therefore a risk of ceiling effects affecting our ability to measure the effect interventions have on empathy. We address this through Study 4.

## Study 4

In Study 4 we modified the stimuli from Study 3 to make the testimonies elicit weaker empathic reactions. We did so by labelling the targets in the stories as legal migrants rather than refugees and changing their stories so that targets reported observing others in difficult situations rather than experiencing them themselves. We kept the Unlimited, Limited and Control interventions from Studies 1-3. In addition, we created a combined intervention based on the previous Malleable and Normative interventions (Combo). This was done because Weisz and colleagues[17] found somewhat stronger effects when a malleability mindset and normativity were combined into one intervention. We also reformulated the Normative intervention in a way that more strongly emphasised empathy's normative nature (See Table SR1 in Supplementary results for a full description).

We recruited 1236 American participants via Prolific. The procedure and measures used were the same as in Study 3, except for modified manipulation checks (see Supplementary Table SR1.3). See Table 1 for descriptive statistics on all measures. Notably, in Study 4, empathic reaction scores were almost a point lower than in Study 3, indicating we had succeeded in lowering overall empathy.

For our fourth study, we hypothesize that interventions Unlimited, Normative and Combo generate more empathy across measures (single-item, reactions, drivers) and donations than Control and that the Limited intervention generates less empathy and prosocial behaviour than Control. We compute contrasts between interventions and interpret evidence as in previous studies.

**Effects of beliefs about empathy being (un)limited.** Participants believed more strongly in the unlimited nature of empathy in the

Unlimited condition than they did in the Control and Limited conditions, but the contrast between the Limited and Control conditions did not exclude zero in the Highest Density Interval (see Supplementary Study 4).

We used the same models as in Study 3, but with adjustments to the priors of the population-level effects of all models, see Methods for model specification. We continued to contrast the Unlimited, Limited and Control conditions on the outcome measures (see Fig. 5). When contrasting Unlimited against Control, we found no differences for single-item empathy ($b_{diff} = -0.12$, $SE = 0.07$, $HDI = [-0.27$ to $0.02]$, $BF_{10} = 0.85$), empathic reactions ($b_{diff} = -0.18$, $SE = 0.07$, $HDI = [0.07$ to $-0.33]$, $BF_{10} = 2.16$) empathic drivers ($b_{diff} = -0.12$, $SE = 0.08$, $HDI = [-0.28 - 0.04]$, $BF_{10} = 0.81$), or, donations ($b_{diff} = -0.02$, $SE = 0.08$, $HDI = [-0.18$ to $0.14]$, $BF_{10} = 0.32$). Participants in the Limited condition did not report lower single-item empathy ($b_{diff} = -0.13$, $SE = 0.07$, $HDI = [-0.28$ to $0.00]$, $BF_{10} = 1.29$) than Control, but they did report lower empathic reactions ($b_{diff} = -0.18$, $SE = 0.07$, $HDI = [-0.33$ to $-0.03]$, $BF_{10} = 4.78$). No difference was found for empathic drivers ($b_{diff} = -0.05$, $SE = 0.08$, $HDI = [-0.22$ to $0.37]$, $BF_{10} = 0.10$), or, donations ($b_{diff} = 0.16$, $SE = 0.08$, $HDI = [-0.00$ to $0.33]$, $BF_{10} = 2.08$). When contrasting Unlimited against Limited, we found no differences for single-item empathy ($b_{diff} = 0.01$ $SE = 0.07$, $HDI = [-0.13$ to $0.16]$, $BF_{10} = 0.27$), empathic reactions ($b_{diff} = 0.02$, $SE = 0.07$, $HDI = [-0.12$ to $-0.17]$, $BF_{10} = 0.27$) or empathic drivers ($b_{diff} = -0.06$, $SE = 0.08$, $HDI = [-0.23$ to $0.10]$, $BF_{10} = 0.41$). Surprisingly, participants in the Unlimited condition donated smaller amounts than participants in the Limited condition ($b_{diff} = -0.18$, $SE = 0.08$, $HDI = [-0.35$ to $0.02]$, $BF_{10} = 3.28$).

**Effects of inducing motivational beliefs about empathy.** Assessing the manipulation checks for our approach-increasing interventions, we found that the beliefs about empathy's normativity did not differ between the Normative and Control conditions. Moreover, the belief about

empathy's malleability did not differ between Combo and Control (see Supplementary Study 4 for all contrasts).

We compared the outcome measures by contrasting Normative and Control, and Combo and Control (Fig. 5). For the contrasts between Normative and Control, no differences were found for single-item empathy ($b_{diff} = -0.01$, $SE = 0.07$, $HDI = [-0.16$ to $0.27]$, $BF_{10} = 0.13$), empathic reactions ($b_{diff} = -0.06$, $SE = 0.07$, $HDI = [-0.21$ to $0.09]$, $BF_{10} = 0.36$), empathic drivers ($b_{diff} = 0.00$, $SE = 0.08$, $HDI = [-0.16$ to $0.17]$, $BF_{10} = 0.3$) or donations ($b_{diff} = 0.11$, $SE = 0.08$, $HDI = [-0.05$ to $0.28]$, $BF_{10} = 0.8$). When the Combo and Control conditions were compared, Control generated more single-item empathy ($b_{diff} = -0.15$, $SE = 0.07$, $HDI = [-0.3$ to $-0.16]$, $BF_{10} = 2.17$) and higher empathic reactions ($b_{diff} = -0.18$, $SE = 0.07$, $HDI = [-0.33$ to $0.03]$, $BF_{10} = 5.72$). We found no differences for empathic drivers ($b_{diff} = -0.05$, $SE = 0.08$, $HDI = [-0.22$ to $0.1]$, $BF_{10} = 0.39$) or donations ($b_{diff} = 0.00$, $SE = 0.08$, $HDI = [-0.15$ to $0.17]$, $BF_{10} = 0.29$).

**Comparing approach and avoidance motives.** Finally, we did not find any difference between the approach-increasing interventions, Normative and Combo, when compared with the avoidance-decreasing intervention, Unlimited (all contrasts are reported in the Supplementary results Study 4, Tables SR5.9, SR5.11, SR5.13, and SR5.15).

**Conclusions Study 4.** To summarize, Study 4 showed that weakening testimonies to (successfully) reduce potential ceiling effects when measuring empathy did not change the pattern of results. Unlimited still does not outperform Control on any measure. Nor did we manage to positively affect empathy through our modified versions of the Malleable or Normative interventions. Thus, we only confirm a small part of our hypotheses, as Limited generated lower empathic reactions than Control, but the rest of the interventions did not generate more empathy and prosocial behaviour than Control.

**Study 5**

Study 5 used a within-subject design so that we could make direct comparisons between in- and outgroup contexts within the same participant and to closely replicate Hasson and colleagues' Study 3, using their precise interventions (Limited, Unlimited and Hasson Control), as well as a modified version of their vignettes. We also retained the Control condition used in studies 1–4 as our Control often performed on par with the active interventions our previous studies. In their Study 3, Hasson and colleagues found that participants in the Unlimited condition did not differ in their empathic reactions or their support for social actions towards in- and outgroup members. In contrast, participants in their Limited and Control conditions had stronger empathic reactions and support for social actions towards ingroup members as compared to outgroup members. Additionally, all participants completed the Interpersonal Reactivity Index[56,57]. This measure was added so that we can examine the effects of adjusting for trait empathy on how participants react to the different interventions.

We recruited 994 American participants via Prolific (498 Democrats; 496 Republicans). This gave us a power of ≥99% to detect the effect size for the target Identity x Condition interaction for empathic reactions found in Hasson and colleagues' Study 3 (Cohen's d = 0.5). See Table 1 for descriptive statistics.

For our final study, we preregistered hypotheses that the Unlimited intervention would generate the most empathy (single-item and reactions) and prosocial behaviour, followed by Hasson Control and then Control. We expected Limited to generate the least amount of empathy and prosocial behaviour among participants. We compute contrasts between interventions and interpret evidence as in previous studies.

**Manipulation checks.** Participants rated their beliefs about empathy being an unlimited resource differently depending on the condition. Participants in the Unlimited condition reported stronger beliefs about empathy being an unlimited resource compared to participants in all other conditions. Participants in the Limited condition believed less in

the unlimited nature of empathy than in the two control conditions (see Supplementary SR6.2).

**Empathic reactions towards in- and outgroup targets within intervention.** To estimate how participants in different conditions rated their empathic tendencies and prosocial support, we included an interaction between condition and stimuli (in- and outgroup targets) in our regression models, see Methods for full model specification. The critical result in Hasson and colleagues' Study 3 was that empathic reactions towards outgroup targets were lower than to ingroup targets in their control and Limited conditions, but that no difference in empathic reactions to in- and outgroup targets was found in the Unlimited condition. In this latter condition, participants' empathic reactions to the outgroup were comparable to their empathic reactions to the ingroup. For ingroup targets, they reported no differences between conditions, hence their effects are driven by an attenuation of an intergroup empathy bias in their unlimited condition.

To test if we obtained the same pattern of results, we compared empathic reactions for in- and outgroup targets. We first examined the Limited condition and found a difference ($bdiff = 0.13$, $SE = 0.06$, $HDI = [0.01–0.25]$, $BF_{10} = 2.5$), such that empathic reactions were higher for ingroup compared to outgroup targets. The same results were obtained for Hasson Control ($bdiff = 0.24$, $SE = 0.06$, $HDI = [0.13–0.37]$, $BF_{10} \geq 10^4$) and our Control condition ($bdiff = 0.29$, $SE = 0.06$, $HDI = [0.17–0.41]$, $BF_{10} \geq 10^4$). Encouragingly, we found no differences in in- and outgroup empathic reactions in the Unlimited condition ($bdiff = 0.08$, $SE = 0.06$, $HDI = [-0.03$ to $0.20]$, $BF_{10} = 0.54$). Prima facie, these findings replicate Hasson's study 3 that the Unlimited condition attenuates an intergroup empathy bias, although the Bayes Factor remains inconclusive. However, this conclusion gets complicated when considering the levels of empathy reported towards the in- and outgroup targets directly (see Fig. 6). Hence, we turn to analyzing these next.

**Comparing interventions' effects on empathic reactions towards in- and outgroup targets.** We started by contrasting empathic reactions toward ingroup targets between the condition Unlimited and the two control conditions, finding no difference excluding null when contrasting against Hasson Control ($bdiff = -0.08$, $SE = 0.08$, $HDI = [-0.24$ to $0.07]$, $BF_{10} = 0.47$) but when contrasting against Control ($bdiff = -0.2$, $SE = 0.08$, $HDI = [-0.36$ to $-0.04]$, $BF_{10} = 5.88$). We also did not find a difference excluding null when contrasting Unlimited against Limited ($bdiff = 0.1$, $SE = 0.08$, $HDI = [-0.05$ to $0.26]$, $BF_{10} = 0.61$). We did find differences when contrasting Limited against both Hasson Control ($bdiff = -0.18$, $SE = 0.08$, $HDI = [-0.34$ to $-0.02]$, $BF_{10} = 3.33$) and against Control ($bdiff = -0.30$, $SE = 0.08$, $HDI = [-0.46$ to $-0.14]$, $BF_{10} = 200$). No difference was found between control conditions. In sum, empathic reactions towards ingroup targets were highest in the two control conditions and lowest in the two intervention conditions.

We continued by looking at empathic reactions toward outgroup targets in the same way as above. We first contrasted condition Unlimited against Hasson Control ($bdiff = 0.08$, $SE = 0.08$, $HDI = [-0.07$ to $0.24]$, $BF_{10} = 0.49$) and Control ($bdiff = 0.01$, $SE = 0.08$, $HDI = [-0.14$ to $0.16]$, $BF_{10} = 0.29$). None of the contrasts excluded null. Neither did we find any differences for the rest of the contrasts, Unlimited against Limited ($bdiff = 0.15$, $SE = 0.08$, $HDI = [-0.00$ to $0.31]$, $BF_{10} = 1.61$), or Limited against Hasson Control ($bdiff = -0.04$, $SE = 0.08$, $HDI = [-0.20$ to $0.12]$, $BF_{10} = 0.31$) or Control ($bdiff = -0.14$, $SE = 0.08$, $HDI = [-0.3$ to $0.01]$, $BF10 = 1.38$). No difference was found between control conditions. Hence, empathic reactions toward outgroup targets were largely similar in all conditions, with only a trend towards lower empathic reactions in the Limited condition, in line with our findings in previous studies.

Taken together, while our results replicate the finding of a reduced intergroup empathy bias in the Unlimited condition compared to other conditions, the result appears to be driven by a different underlying

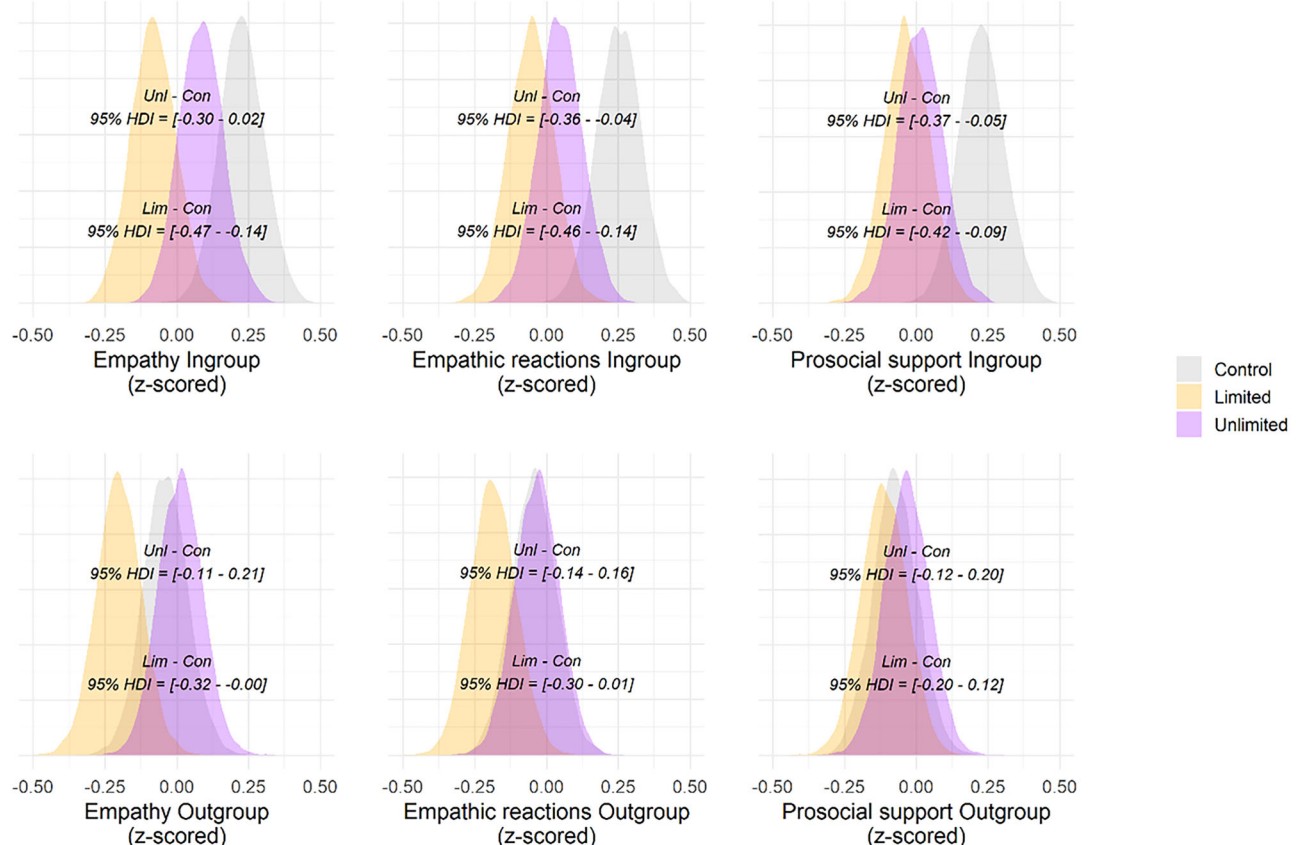

**Fig. 6 | Density distributions of empathy measures and support for social action from Study 5.** Density distributions (all z-scored) from multiple regression models for single-item empathy, empathic reactions, and support for prosocial action for in- and outgroups. The first row shows results for the ingroup, with the bottom HDI interval showing the contrast between Limited and Control for, while the top HDI interval shows the contrast between Unlimited and Control. The second row shows results for the outgroup, with the bottom HDI showing the contrast between Limited and Control, and the top HDI shows the contrast between Unlimited and Control. Figure based on participants from three conditions: Unlimited = 249; Limited = 246; Control = 249.

pattern of responses compared to Hasson and colleagues' results. Importantly, in our results, the effect is not driven by participants in the Unlimited condition having higher empathic reactions toward either in- or outgroup targets compared to the rest of the conditions, but rather by the rest of the conditions generating stronger empathic reactions towards ingroup targets.

**Single item empathy.** We also found a tendency for single item empathy being lower for ingroup targets in all contrasts against Limited: Control ($b_{diff} = -0.30$, $SE = 0.08$, $HDI = [-0.47$ to $-0.14]$, $BF_{10} = 333.33$) and Hasson Control ($b_{diff} = -0.19$, $SE = 0.08$, $HDI = [-0.35$ to $-0.03]$, $BF_{10} = 4$) and Unlimited condition ($b_{diff} = -0.16$, $SE = 0.08$, $HDI = [-0.04$ to $-0.33]$, $BF_{10} = 2$). This pattern repeated itself for outgroup targets for contrasts Limited against Control ($b_{diff} = -0.16$, $SE = 0.08$, $HDI = [-0.32$ to $-0.00]$, $BF_{10} = 2.04$) and against Unlimited ($b_{diff} = -0.21$, $SE = 0.08$, $HDI = [-0.04$ to $-0.37]$, $BF_{10} = 7.69$). None of the other contrasts for single-item empathy excluded null (see Fig. 6; supplementary Table 6.6 for full results). Once again, when encountering brief interventions, participants appear to be more prone to down-regulating than to upregulating their empathy.

**Support for prosocial action.** For the compound measure "support for prosocial action", we found that participants in condition Control support prosocial actions more than participants in Unlimited ($b_{diff} = 0.21$, $SE = 0.08$, $HDI = [0.37–0.05]$, $BF_{10} = 7.69$) and Limited ($b_{diff} = 0.26$, $SE = 0.08$, $HDI = [0.42–0.09]$, $BF_{10} = 100$) when encountering outgroup targets. No other contrasts excluded the null (see Fig. 6; and supplementary Table SR6.16 for full results).

**Adjusting for variation in empathic traits.** Finally, we used the four subscales of the Interpersonal Reactivity Index (IRI) to explore the effects of adjusting for variation in participants' empathic traits for evaluating the interventions. Interestingly, when we adjusted for the influence of personal distress, which captures self-oriented feelings of anxiety, we found that empathic reactions in the Limited condition were reliably lower compared to Control ($b_{diff} = -0.24$, $SE = 0.07$, $HDI = [-0.38$ to $-0.10]$, $BF_{10} = 67$) and Unlimited ($b_{diff} = -0.17$, $SE = 0.07$, $HDI = [-0.31$ to $-0.02]$, $BF_{10} = 3.5$) and somewhat lower compared to Hasson Control ($b_{diff} = -0.14$, $SE = 0.07$, $HDI = [-0.28$ to $0.001]$, $BF_{10} = 1.81$). A similar pattern of results obtained for single-item empathy and prosocial action (see supplementary Table SR6.24 – SR6.26). Adjusting for the three other IRI subscales: perspective taking, fantasy and empathic concern, we extended some of the patterns but not as strongly as for personal distress (see supplementary Tables SR6.18-6.23 and SR6.29 for full results). To conclude, adjusting for empathic traits appeared to accentuate differences brought about by the effects of the Limited condition, overall lowering participants' empathic reactions and prosocial support.

**Conclusions Study 5.** In sum, in Study 5, we replicate the results from Hasson and colleagues' Study 3, in that we find similar empathic reactions towards in- and outgroup targets for participants in the condition Unlimited. However, when investigating the underlying patterns that create this result, we find small differences between the other conditions (Limited, Control, Control Hasson) that, when contrasted against each other, result in different empathic reactions between target groups. The differences between empathic reactions to in- and outgroup targets are smaller for condition Unlimited than for the other conditions, but

Unlimited does not generate stronger empathic reactions towards either in- or outgroup than the two control conditions do. We only partly confirmed our hypotheses as intervention Limited for some measures generated less empathy than the other interventions, but the Unlimited did not generate more empathy than controls.

## Discussion

We tested whether brief motivated empathy interventions could make participants report more empathy and behave more prosocially towards outgroup members. We attempted to replicate and generalize previously reported successful interventions[18] as well as test new interventions based on existing theoretical models[1,17,47]. Despite promising reports in the literature[18], our results indicate that brief interventions fail to reliably raise empathy or promote prosocial behaviour. In what follows, we first discuss our attempted replication of Hasson and colleagues' interventions and the suggestion that beliefs regarding the limited nature of empathy affect how we empathize with outgroup members. We then discuss interventions inspired by Weisz and colleagues, and compare interventions based on approach-increasing and avoidance-decreasing motives. Finally, we briefly discuss our studies' limitations and summarise our main contributions to the study of motivated empathy.

Our results show that inducing the belief that empathy is an unlimited resource had no statistically reliable effect in motivating participants to feel empathy. While we consistently failed to observe any differences in empathy between interventions and control, we only partly found statistical support for the null. The Unlimited intervention did not perform better than a control intervention, regardless of whether the outcome was measured as a single-item, or as empathic reactions, or as prosocial behaviour (Studies 1–5), even when we used the same set of stimuli as in the original study (Study 3). Importantly, in Study 5, we also included a second control condition, replicated from Hasson and colleagues' Study 3. We found no difference between their control and the control we have been using throughout our studies on any measure or in any context.

It is important to ask if a failure to replicate prior results stems from methodological problems. The failure to increase empathy is not because the Unlimited intervention failed to induce appropriate beliefs. Participants exposed to it all believed more strongly that empathy is not a limited resource. The suitable pattern of beliefs held for comparisons against the Limited condition in all studies and for comparison to control conditions in all studies but Study 2. Additionally, in Study 4, we controlled for potential ceiling effects that otherwise could have affected our ability to observe differences in empathy. In Study 5, we used a within-participants design to exclude potential between-participants variability that might affect interventions and our results. Still, participants in the Unlimited condition did not report more empathy, empathic reactions or support for prosocial actions than participants in the two control conditions. In addition, those exposed to the Unlimited intervention did not perform better than the Control for the auxiliary measures of empathic drivers and monetary donations. In Study 3, the only case where Unlimited's empathic drivers did differ positively from the Control, we found that inducing the opposite belief – that empathy is a limited resource – also increased empathic drivers, limiting the weight of this result. Taken together, we argue that the failure to replicate that inducing beliefs about empathy being unlimited increases reported empathic tendencies or closes intergroup empathy gaps is unlikely to stem from methodological failures to implement the intervention.

How can we understand our results in relation to the theoretical framework proposed by Hasson and colleagues? They suggest that people's beliefs about empathy being a limited resource play a role in shaping intergroup empathy bias. Their claim is that when people think of empathy as a limited resource, they tend to keep that resource for their ingroup; when they think of it as an unlimited resource, they are more inclined to share their empathy with both in- and outgroup members. Our results indicate that people's beliefs about the limited nature of empathy do not have the suggested effects, but also that those beliefs are not entirely inert. Participants in

the Limited condition showed weaker empathic reactions than the Control conditions in two of our five studies, with an additional study excluding null in the HDI, whereas Unlimited *never* outperformed the Control conditions. This suggests that informing participants that empathy is a limited resource potentially makes them *less* empathic, but we never found support for that informing participants that empathy is an unlimited resource makes them *more* empathic. It is therefore unlikely that beliefs about empathy *not* being a limited resource reduce the intergroup empathy bias. This highlights the difficulty of constructing minimal motivated empathy interventions that get people to upregulate empathy.

Our results indicate that beliefs about empathy affect parts of participants' motivation to empathize. While the Limited intervention did not consistently decrease empathy across all studies, it may nevertheless be useful in future research on intergroup empathy. If we find a way to induce beliefs about the limited nature of empathy that are reliably coupled with lower empathic reactions, this intervention could be used to further scrutinize different drivers of the intergroup empathy bias[13,62–66]. Thus, the Limited intervention may function as a vehicle to test competing causal accounts of the roots of intergroup empathy bias. As the Limited intervention is brief, it can be applied in a variety of contexts and to populations that cannot be treated with longer interventions.

In study 5, we additionally explored the effects of adjusting for participants' trait empathy using the IRI. We found that the effect of Limited became stronger when controlling for the IRI personal distress subscale. Some previous research has demonstrated that distress can hamper empathic tendencies and prosocial behaviour[67–72]. We consider this result to increase the credibility of the claim that Limited makes participants downregulate their empathic tendencies, as Limited has an even stronger effect when controlling for other influences that can lower empathy. Nonetheless, the relationship between state and trait empathy appears convoluted. A recent study found that trait empathy only explains a limited amount of the variance in the daily experience of empathy[73]. We caution against overinterpreting any effects that trait empathy has on state empathy when these are amplified through interventions.

We next consider the other interventions tested in this paper. In contrast to the interventions proposed by Hasson and colleagues, they targeted approach motivations: beliefs that we can increase our empathy, that empathy is socially desirable or normative, or a combination of the two. However, no approach intervention increased empathy or empathic reactions above the Control. Similarly, they had no consistent effect on empathic drivers or donations. Since the beliefs involved are complex, brief interventions may not be appropriate for saliently conveying the relevance of empathy's malleability and normativity by actualizing intended motivations among participants. Weisz and colleagues reported some success in making participants more empathetic and prosocial by using longer interventions targeting the same beliefs. It may be that a long format is necessary to make participants properly internalize the content of these interventions.

A third goal of our studies was to examine how the efficiency of either increasing the approach to or decreasing the avoidance of empathy is affected by an in- or intergroup context[1,17,47]. We compared how the interventions fared in relation to one another in each context (intergroup context Studies 1, 3, 4 and ingroup context Study 2). We found no noteworthy difference between approach-increasing and avoidance-decreasing interventions when meeting in- or outgroup targets. Unfortunately, since none of the interventions managed to positively affect participants' empathy to begin with, it is difficult to draw firm conclusions. Still, this is the first formal test between approach-increasing and avoidance-decreasing interventions applied in both an in- and intergroup context. Future research should continue this work, ideally with longer interventions that can consistently provide the desired effects. Another interesting avenue could be to design interventions that combine both types of motives, e.g., first decreasing motivation to avoid empathy and then trying to get people to approach it and examine how these kinds of interventions perform in intergroup contexts.

## Limitations

Our studies have several strengths, including large sample sizes with high statistical power and combine both exact and conceptual replications of previous work. Nevertheless, they are not without limitations. One main limitation, as already discussed, is that the Normative and Malleable interventions failed to shift participants' beliefs. There are likely additional alternative formulations of these interventions targeting the relevant underlying motives that future work can assess.

A second limitation, which is not unique to this work, is the reliance on brief self-report measures to assess empathy. While self-report measures are practical and ubiquitous, they are likely not sensitive enough to give a full picture of participants' empathic tendencies. It may therefore be possible that brief interventions can show results using alternative outcome measures, such as empathic accuracy tasks or physiological readings. We combined self-reported measures with behavioural measures by letting participants make real monetary donations (study 1–4). That gave us more leverage to find a potential heightening effect of interventions on participants' empathic tendencies and prosocial behaviour, but no effect was found.

A third limitation concerns the use of online experiments in all our studies. This experimental context might potentially affect interventions in unforeseen ways, for example, by diminishing how effective interventions are. However, the studies by Hasson and colleagues, which we attempted to replicate (Studies 2 and 3), also relied on an online sample[18]. We viewed this applicability to online life, an area where information is easily distorted and emotions are amplified[74,75], as a particularly promising feature of their interventions. Interventions that are brief and can be administered via screens allow for additional flexibility and scalability without diminishing their potency. However, we acknowledge that we cannot rule out that some critical differences between online and offline samples remain in how empathy is regulated, for example, through the presence of additional interpersonal cues afforded by typical offline interaction.

One final, important limitation concerns potential covariates. It is well-known that social context and participants' characteristics affect the effectiveness of intergroup interventions[76]. Possibly, our interventions would have been more effective if implemented in contexts with highly salient group boundaries, e.g., an intergroup situation comprising Israelis and Palestinians (however, in our Studies 3 and 4, we used the same type of participants as Hasson and colleagues did in their Study 2, but did not obtain similar results). Likewise, we cannot rule out any cross-cultural differences accruing from our use of US American samples instead of Israeli samples. Additionally, the interventions might work differently depending on participants' individual differences. In Study 5, we controlled for the potential influence of participants' trait empathy on interventions and mainly found that when controlling for personal distress, Limited lowered participants' empathic tendencies further. Future research should continue to determine the boundary conditions for both interventions and motivations by addressing potential covariates.

## Conclusion

There are good reasons to believe that empathy is partly a motivated process[1,25], but this does not entail that empathy is easily shifted. We have shown that several brief, motivated empathy interventions fail to robustly and reliably affect participants' empathy, using several measures. Our findings cast doubts on whether it is possible to induce empathy by making participants believe in its unlimited nature. On the other hand, making participants believe that empathy is limited often reduced reported empathy, suggesting that it might be easier to get people to decrease empathy than it is to get them to increase it. Although beliefs about empathy being limitless do not appear to affect empathy, beliefs about empathy may still affect participants' tendency to empathize.

## Data availability

Data needed to replicate the analysis are available at: https://osf.io/hr6gn/overview.

## Code availability

scripts needed for running the experiments and analysing the data are available at: https://osf.io/hr6gn/overview.

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

## Acknowledgements

A.T. received funding for this study from the Hultengren Foundation and Lund University's Agenda 2030 graduate school. PP was funded by the Swedish Research Council grant 2020-02584. The funders had no role in study design, data collection and analysis, decision to publish or preparation of the manuscript. The authors thank Eran Halperin and Yossi Hasson for upholding open science practices by generously sharing the stimuli used in their second study.

## Author contributions

A.T. proposed the study idea. A.T., A.W. & P.P. contributed to the study design. A.T. collected data and conducted analyses with assistance from P.P. A.T. wrote the paper while A.W. & P.P. provided critical revisions. A.W. & P.P. jointly supervised this work.

## Funding

## Competing interests

The authors declare no competing interests.
