## [Transparent Peer Review file · Communications Psychology]

Brief empathy interventions can decrease but not increase empathic tendencies

Corresponding Author: Mr Alexander Tagesson

Version 0:

Decision Letter:

Dear Mr Tagesson,

Your manuscript titled "Brief empathy interventions can decrease but not increase empathic tendencies" has now been seen by one of the previous reviewers, who previously had outstanding concerns. Their comments appear below. In light of their advice I am delighted to say that we are happy, in principle, to publish a suitably revised version in Communications Psychology.

We also commissioned signed comments from the authors of the target paper, which are also attached for your information. Please note that signed comments are not reviews and need not be replied to in a rebuttal.

We therefore invite you to revise your paper one last time to address the remaining concerns of our reviewers and a list of editorial requests. At the same time we ask that you edit your manuscript to comply with our format requirements and to maximise the accessibility and therefore the impact of your work.

EDITORIAL REQUESTS:

In particular, I highlight two issues:

The first is a request to clarify the use of Bayesian statistics (the Savage–Dickey method) and ensure that the reported BFs are only interpreted if they meet appropriate standards of strength of evidence.

The second request is to not only link to the comprehensive preregistrations (which are commendable), but to also ensure the reporting of analyses and hypotheses can be easily mapped between the preregistration and the paper.

SUBMISSION INFORMATION:

OPEN ACCESS:

At acceptance, you will be provided with instructions for completing the open access licence agreement on behalf of all authors. This grants us the necessary permissions to publish your paper. Additionally, you will be asked to declare that all required third party permissions have been obtained, and to provide billing information in order to pay the article-processing

charge (APC).

* **DATA AVAILABILITY:**

Link Redacted

Best regards,

Marike

Marike Schiffer, PhD
Chief Editor
Communications Psychology

REVIEWERS' COMMENTS:

Reviewer #1 (Remarks to the Author):

The revised manuscript shows clear improvements. However, it is important to emphasize that this does not constitute a true replication, since all of the studies were conducted online. I recommend that the authors explicitly highlight this limitation in the manuscript.

Signed Comments for Replication Study COMMSPSYCHOL-25-0640-T

We thank the authors for their attempt to replicate our work. Replication efforts are essential for advancing cumulative science, and we appreciate the careful attention given to our methodology and findings.

A direct comparison of ingroup and outgroup empathy within the same participants, as in Hasson et al. (2022), was expected in all studies but was conducted only once out of five. Such a within-participant approach controls for individual differences and provides stronger evidence for reducing intergroup empathy bias. In contrast, reliance on between-participant comparisons introduces variability across participants that may obscure true effects.

Given the between-participant design of most studies (four out of five), it is essential to measure and control for participants' baseline empathy or general motivated empathy traits. Without such controls, it is difficult to determine whether observed or null effects result from the interventions themselves or from pre-existing differences between conditions.

In four out of five studies, the control condition differed from that used in the original paper and involved risk priming, which may itself elicit negative affect and place participants in a non-neutral state. Without a neutral control baseline, as operationalized in the original paper, direct comparisons across studies are difficult, particularly when interpreting null effects and evaluating the true impact of the interventions.

Importantly, the replication provides two convergent indications that beliefs about the limitation of empathy are conceptually linked to empathy bias. The findings in Study 1 show that endorsing the view that empathy is limited led to lower empathic responses, suggesting an underlying relationship between such beliefs and empathic processes.

In addition, Study 5, which was the only one employing a within-participant design and a neutral control condition, showed that in the unlimited empathy intervention, no intergroup empathy bias was observed, similar to the results of the original paper. Although the specific comparisons between conditions did not fully align with those reported in the original paper, it is encouraging to see that the closer the study design was to the original, the more the replication converged with those results.

We hope that future studies will further clarify the mechanisms of this intervention and its potential to increase empathy.

Sincerely,

Yossi Hasson, PhD

The Hebrew University of Jerusalem & Reichman University